# Loss of a major toxin gene cluster defines a metabolic schism and host-specific virulence in *Botrytis pseudocinerea*

Victor Coca-Ruiz[1,2,3], Adrián García-Barba[2], Josefina Aleu[2,3], Isidro G. Collado[2,3*]

1 Instituto de Hortofruticultura Subtropical y Mediterránea "La Mayora" (IHSM), CSIC-UMA, Campus de Teatinos, Málaga, Spain, 2 Departamento de Química Orgánica, Facultad de Ciencias, Universidad de Cádiz, Cádiz, Spain, 3 Instituto de Investigación en Biomoléculas (INBIO), Universidad de Cádiz, Cádiz, Spain

* isidro.gonzalez@uca.es

## Abstract

*Botrytis pseudocinerea* is a cryptic fungal species, sympatric with the notorious plant pathogen *Botrytis cinerea*, yet possessing distinct ecological traits including intrinsic fungicide resistance. Despite this advantage, *B. pseudocinerea* rarely dominates agricultural ecosystems, presenting an ecological paradox. This study resolves this paradox by defining the unique pathogenic identity of *B. pseudocinerea* isolate VD165. We demonstrate that VD165 exhibits superior vegetative growth and stress tolerance compared to *B. cinerea* B05.10, coupled with heightened virulence on solanaceous hosts (tomato, tobacco) but reduced virulence on grape. A comprehensive bio-guided chemical investigation reveals a fundamental metabolic schism: the constitutive and infection-induced upregulation of botcinin polyketides (via *Bcboa6/B-cboa9*) contrasted with the complete functional loss of the botrydial sesquiterpene pathway. This loss is biochemically confirmed by the significant accumulation of the upstream precursor mevalonolactone. This chemotype, loss of botrydial and compensatory super-activation of botcinins, phenocopies *B. cinerea* ΔBcbot2 mutants, establishing *B. pseudocinerea* as a "natural knockout" model that validates the inverse regulation of these major toxin families. We propose that this "evolution by subtraction" is a lineage-specific adaptation shared with the sister species *B. fabae*, driving host specialization and defining the ecological niche of *B. pseudocinerea*.

## Introduction

The genus *Botrytis* contains some of the most significant plant pathogens known to agriculture. Foremost among them is *Botrytis cinerea*, the necrotrophic fungus responsible for grey mould disease. This pathogen is a global agricultural scourge, capable of infecting more than 1400 plant species, including a vast array of economically vital fruit, vegetable, and ornamental crops [1]. The economic devastation

**Data availability statement:** All relevant data are within the manuscript and its Supporting information files.

**Funding:** This work was supported by the MICIU/AEI under Grant PID-2021-122899 OB-C21 (10.13039/501100011033) and by ERDF/EU.

**Competing interests:** The authors have declared that no competing interests exist.

wrought by *B. cinerea* is immense, with annual worldwide losses conservatively estimated to be between $10 billion and $100 billion [2–4]. In specific sectors, such as tomato production, grey mould can lead to catastrophic yield reductions of over 80%, impacting crops both pre- and post-harvest [1,2].

For decades, control of *B. cinerea* has relied heavily on the application of chemical fungicides, a market segment that alone accounts for approximately 8% of total global fungicide sales [1,3]. However, this strategy is increasingly compromised by the rapid evolution of fungicide-resistant strains and growing public and regulatory pressure to reduce chemical inputs in agriculture due to environmental and health concerns [5,6]. This precarious situation highlights an urgent need for more sustainable and biologically informed disease management strategies, which can only be developed through a deeper understanding of the pathogen's biology and diversity.

A significant complication in understanding and managing fungal pathogens is the phenomenon of cryptic speciation, where two or more distinct species are morphologically identical but are, in fact, genetically divergent and reproductively isolated [7]. Such sibling species can exhibit significant differences in their ecology, host preference, virulence, and sensitivity to control agents, yet remain indistinguishable by traditional visual identification [8]. The failure to recognise this hidden diversity can lead to flawed epidemiological models and ineffective disease control measures. The field of mycology is replete with examples where molecular phylogenetics has unmasked cryptic species complexes within supposedly single pathogenic entities, such as in the genera *Aspergillus*, *Paracoccidioides*, and *Histoplasma*, fundamentally altering our understanding of these diseases [7].

Within the *Botrytis* genus, such a case of cryptic speciation was definitively established by Walker *et al.* (2011) [9], who formally delineated *Botrytis pseudocinerea* as a species distinct from *B. cinerea* [10]. This reclassification was not based on subtle variations but on solid, multi-faceted evidence. Phylogenetic analyses of multiple gene genealogies revealed that the two species belonged to different, well-supported phylogenetic clades. None of the morphological criteria tested (spore size, germination rate, or mycelial growth) were able to discriminate between these two species. Sexual crosses between individuals from the same species and different species were carried out. Only crosses between individuals from the same species were successful. Moreover, population genetics analysis revealed a high level of diversity within each species and a lack of gene flow between them [8]. The evolutionary divergence between these lineages is striking, estimated to have occurred 7–18 million years ago, with *Botrytis fabae* being a closer relative to *B. pseudocinerea* than *B. cinerea* is [9,11]. Crucially, genomic studies have indicated that *B. fabae* also lacks a functional botrydial biosynthetic cluster, suggesting that this loss is likely a lineage-specific trait shared across Clade 1 rather than an isolate-specific anomaly [12].

Despite this deep genetic chasm, the two species are morphologically indistinguishable and frequently co-occur in the same agricultural niches, creating a classic cryptic species complex [9]. Phylogenetically, *B. pseudocinerea* resides in Clade 1, a lineage distinct from the *B. cinerea* complex (Clade 2) and shared with

*Botrytis fabae.* Unlike the generalist *B. cinerea*, members of Clade 1 often exhibit narrower host ranges or specialized virulence strategies. This species has since been identified as an emerging pathogen on a global scale, causing grey mould on crops such as tomato in China [13], blueberry in North America [14], and *Brassica napus* in China [15]. This coexistence presents a compelling ecological paradox [8]. *B. pseudocinerea* possesses an intrinsic, genetically determined resistance to fenhexamid, a fungicide widely used to control grey mould [8,9]. Standard evolutionary theory would predict that in environments where fenhexamid is applied, *B. pseudocinerea* should gain a significant competitive advantage and eventually dominate the pathogen population. However, field studies have observed the contrary; *B. cinerea* often displaces *B. pseudocinerea* in treated fields, indicating that factors beyond simple fungicide resistance are at play in determining ecological fitness and pathogenic success [8].

The pathogenic success of necrotrophic fungi like *Botrytis* is largely determined by their profile of phytotoxic secondary, particularly the secretion of phytotoxic secondary metabolites that kill host cells and facilitate colonization [16,17]. The metabolome of *B. cinerea* is well-characterised and known to be dominated by two families of phytotoxins: the sesquiterpenoid botrydial and the polyketide botcinic acid (**1**) [16–18]. These compounds collectively induce chlorosis and cell collapse, believed to aid in the invasion and colonization of plant tissue [19]. Botrydial, in particular, is considered a pivotal pathogenicity factor, consistently detected during the infection of plant tissues [20]. To date, the secondary metabolite profile of *B. pseudocinerea* remains entirely unknown.

While pathogen evolution is often associated with the acquisition of new virulence factors, the loss of specific functions can be an equally potent driver of adaptation and speciation. This process, sometimes termed "evolution by subtraction," can lead to niche specialization [21]. We hypothesised that the distinct ecological behaviour and host-dependent virulence of *B. pseudocinerea* are underpinned by a major evolutionary event: a metabolic schism involving the complete functional loss of the botrydial secondary metabolite pathway. We posited that this loss represents a 'natural knockout' event, offering an unique opportunity to study toxin regulation without the artifacts of genetic engineering. By characterizing the metabolome and physiology of *B. pseudocinerea* VD165, we aim to test the hypothesis that this species has undergone a 'metabolic schism', the evolutionary shedding of the botrydial pathway, as a trade-off for specialized virulence, mirroring the specific toxin regulation mechanisms previously elucidated only through artificial gene deletion.

## Materials and methods

### Fungal strains, plant material, and growth conditions

The *Botrytis pseudocinerea* strain VD165, isolated from grapevine in France, was generously provided by Dr. Muriel Viaud (INRAE, France) and serves as the reference genotype for this cryptic species in previous population and genomic studies [8,9]. The well-characterised reference strain *Botrytis cinerea* B05.10 was used for all comparative studies [22]. Both strains were routinely maintained and cultured on YGG-agar medium, consisting of 2% (w/v) glucose, 0.5% (w/v) yeast extract, and 0.3% (w/v) Gamborg B5 basal medium with vitamins (Duchefa Biochemie, Haarlem, The Netherlands; ref. G0209.0005), solidified with 1.5% (w/v) agar. For qualitative assessment of medium acidification, malt agar medium (Condalab, Madrid, Spain; ref.1038) was used. All cultures were incubated at a constant temperature of 20°C. For long-term preservation, mycelial stock suspensions were prepared in 10% (v/v) sterile glycerol and stored at −80°C. Plant materials for virulence assays, including petals of *Gerbera jamesonii*, fruits of grape (*Vitis vinifera*), and tomato (*Solanum lycopersicum*), were sourced from local markets and ensured to be free of pesticides and visible damage. *Nicotiana tabacum* cv. Havana and *Solanum lycopersicum* cv. Moneymaker plants were cultivated from seed in a controlled environment growth chamber (Sanyo MLR-352H) set at 22°C with 70% relative humidity. A photoperiod of 14 hours of light and 10 hours of dark was maintained using cool white fluorescent lamps providing a light intensity of approximately 150 µmol m$^{-2}$s$^{-1}$.

## Vegetative growth and stress tolerance

To assess radial growth on solid medium, 90-mm Petri dishes containing 25 mL of YGG-agar were centrally inoculated with a 5-mm diameter mycelial agar plug taken from the actively growing margin of a 3-day-old colony. The colony diameter was measured daily in two perpendicular directions for four consecutive days, and the radius was calculated. The results presented are the mean values from fifty biological replicates ($n = 50$) ± standard deviation.

For stress sensitivity assays, YGG-agar medium was supplemented post-autoclaving with one of the following sterile-filtered stress agents to the final concentrations indicated: 1 M Sorbitol for osmotic stress (Sigma-Aldrich, St. Louis, MO, USA; ref. S1876), 5 mM hydrogen peroxide ($H_2O_2$) for oxidative stress (Foret, Huelva, Spain; ref. 018556), or 0.02% (w/v) sodium dodecyl sulfate (SDS) for cell wall stress (Sigma-Aldrich; ref. L3771). Plates were inoculated as described above. The radial mycelial growth radius ($R$, in cm) was measured at 72 h for both treated ($Rt$) and control ($Rc$) plates. The percentage of growth inhibition (GI) was calculated using the formula: GI(%)=×100.

For biomass evaluation in liquid culture, 100-mL Erlenmeyer flasks containing 30 mL of liquid YGG medium were inoculated with five 5-mm mycelium agar plugs. The flasks were incubated for four days at 20°C with gentle orbital shaking (40 rpm) in an Infors HT Multitron shaker. Mycelia were harvested by filtration through Miracloth (Calbiochem, San Diego, CA, USA), washed twice with sterile distilled water, and blotted dry on filter paper to determine fresh weight. Subsequently, the mycelia were dried in an oven at 50°C to a constant weight for dry weight determination. This experiment was conducted with 45 biological replicates ($n = 45$).

## Assessment of virulence

Virulence assays were conducted on detached plant organs to assess pathogenic potential. Fully expanded young leaves of tobacco and tomato, and fresh gerbera petals were placed on 1% water agar in sealed 150-mm Petri dishes to maintain high humidity. Each sample was inoculated with a 5-mm diameter mycelial plug taken from a YGG-agar culture. To facilitate infection, the plug was placed on the plant surface along with a 20 μL droplet of sterile 0.5% (w/v) Potato Dextrose Broth (PDB) (Condalab; ref.1261). The plates were incubated in darkness at 20°C. Lesion diameters were quantified at 72 hours post-inoculation (hpi) using ImageJ software (v1.53) [23]. The results are presented as the mean value from one hundred biological replicates ($n = 100$).

Fungal virulence on fruits was evaluated using a semi-quantitative scale based on the progression of rot. Tomato and grape fruits were surface-sterilised, wounded with a sterile needle, and inoculated as described for leaves. Infection was scored daily according to four stages: Stage 1 (initial maceration at the inoculation site), Stage 2 (<25% of fruit surface covered by rot), Stage 3 (25–75% coverage), and Stage 4 (>75% coverage with visible sporulation). Results are presented as the percentage of fruits in each stage at the final time point, derived from 100 biological replicates ($n = 100$).

## Evaluation of reactive oxygen species (ROS) and medium acidification

To assess endogenous ROS production, mycelia were grown for three days on YGG-agar plates overlaid with a sterile cellophane membrane. Quantitative determination of H2 O2 for thr was performed using a 3,3-Diaminobenzidine (DAB) assay. Twenty-four milligrams of fresh mycelium were harvested from the cellophane, transferred to a microfuge tube, and incubated with 1 mL of a 10 mg/mL DAB solution (Sigma-Aldrich; ref. D7304) in 10 mM MES buffer (pH 6.5) for 5 h in darkness with gentle agitation. The reaction was stopped by the addition of 1 M HCl. The absorbance of the brown polymerised DAB product in the supernatant was measured at 471 nm using a BioTek Synergy HT spectrophotometer. Results are presented as the mean value from 50 biological replicates ($n = 50$).

For qualitative analysis of medium acidification, malt-agar plates were supplemented with 0.01% (w/v) bromocresol green as a pH indicator (Schumacher *et al.*, 2013). The plates were centrally inoculated and incubated for up to 7 days. Acidification, indicative of organic acid secretion (e.g., oxalic acid), was visualised by a colour change of the indicator from blue (pH > 5.4) to yellow (pH < 3.8).

  

### Rationale for bio-guided fractionation and identification of secondary metabolites

The onion (*Allium cepa*) epidermis plasmolysis assay was selected as a rapid, cost-effective, and biologically relevant screen for phytotoxicity. This assay directly measures membrane damage and cell death, which are key effects of necro-trophic phytotoxins, making it an ideal 'bio-guide' to prioritize fractions containing active compounds for intensive chemical characterization, as this bioassay directly measures the phytotoxic activity central to the infection mechanism [19,24]. For initial screening, *B. pseudocinerea* VD165 was grown in 50 mL of modified Czapek-Dox liquid medium in 250-mL flasks for 7, 14, and 21 days, under both continuous light and continuous darkness. The culture broth from each condition was extracted with an equal volume of ethyl acetate, and the organic extracts were concentrated. The phytotoxicity of each crude extract was assessed to identify the most potent culture condition [24].

Based on the screening results, large-scale production was performed. Forty 250-mL Erlenmeyer flasks, each containing 130 mL of modified Czapek-Dox medium, were inoculated with five mycelial plugs of VD165. The flasks were incubated for 14 days at 20°C with orbital agitation (140 rpm) under continuous light. The total culture filtrate (5.2 L) was pooled and extracted three times with equal volumes of ethyl acetate. The combined organic layers were dried over anhydrous $Na_2SO_4$ and concentrated under reduced pressure to yield 590 mg of crude extract. This extract was subjected to silica gel column chromatography (230–400 mesh) using a step gradient of n-hexane-ethyl acetate followed by ethyl acetate-methanol to afford several fractions. Fractions containing compounds of interest were further purified by semi-preparative High-Performance Liquid Chromatography (HPLC) on an Agilent 1260 Infinity II system equipped with a Zorbax Eclipse XDB-C18 column (9.4 x 250 mm, 5 µm). Isocratic or gradient elution with mixtures of acetonitrile/water or methanol/water, containing 0.1% formic acid, at a flow rate of 3 mL/min was used to isolate pure compounds, with detection at 220 nm and 254 nm.

### Spectroscopic characterisation and gene expression analysis (qRT-PCR)

The chemical structures of all isolated compounds were elucidated by a combination of spectroscopic methods. Nuclear Magnetic Resonance (NMR) spectra ($^1$H, $^{13}$C, COSY, HSQC, HMBC) were recorded on Bruker 400, 500, and 700 MHz spectrometers. High-Resolution Mass Spectrometry (HRMS) data were obtained on a Waters Xevo G2-XS Q-TOF mass spectrometer with an electrospray ionisation (ESI) source in negative mode. Infrared (IR) spectra were also recorded. For HRMS acquisition, the spectrometer operated with a capillary voltage of 2.5 kV, a sampling cone voltage of 40 V, and a source temperature of 120°C, acquiring data over a mass range of *m*/*z* 50–1200.

For gene expression analysis, total RNA was extracted from mycelia grown in axenic YGG culture (7 and 14 days) and from infected tobacco and tomato leaf tissues (24 and 48 hpi) that were flash-frozen in liquid nitrogen. Samples were homogenised in Trizol Reagent (Invitrogen, Carlsbad, CA, USA) according to the manufacturer's protocol [25]. As the *B. pseudocinerea* VD165 genome is not annotated [12], primers for qRT-PCR were designed based on conserved sequences. The sequences of *B. cinerea* B05.10 botcinic acid biosynthesis genes (*Bcboa6*/*Bcpks6*, *Bcboa9*/*Bcpks9*) were used in a BLASTn search (Camacho *et al.*, 2009) against the genome of *B. pseudocinerea* BP362 (GCA_019395245.1) and other related species (Valero-Jiménez *et al.*, 2019). Homologous sequences were aligned using Clustal Omega to identify conserved regions for primer design (Table 1). qRT-PCR was performed on a Bio-Rad CFX96 Real-Time PCR Detection System using iQ SYBR Green Supermix (Bio-Rad). Relative mRNA levels were quantified using the $2^{-\Delta\Delta Ct}$ method, normalised to the geometric mean of two reference genes, actin (*bcactA*) and β-tubulin (*bcβtub*), and expressed relative to the expression level in *B. cinerea* B05.10 under the same condition [25].

### Statistical analysis

All statistical analyses were performed with GraphPad Prism 8 software (GraphPad Software, San Diego, CA, USA). Data sets were first tested for normality using the Kolmogorov-Smirnov or Shapiro-Wilk test. For comparisons between two groups of normally distributed data, an unpaired, two-tailed Student's t-test was used. For data that were not normally

**Table 1. Primers used for qRT-PCR in this study.**

| Name | Sequence | Description |
|---|---|---|
| Fw*bcboa6* | GGTGAGGTTGAAGTGCCG | mRNA level quantification of the gene *Bcboa6*/*Bcpks6* |
| Rv*bcboa6* | TGCCTGCAGGATAAGCTC | mRNA level quantification of the gene *Bcboa6*/*Bcpks6* |
| Fw*bcboa9* | TCGTCACATGGTTGAAAAGG | mRNA level quantification of the gene *Bcboa9*/*Bcpks9* |
| Rv*bcboa9* | TGCTTTTTGGACGTCGGA | mRNA level quantification of the gene *Bcboa9*/*Bcpks9* |
| Fw*bcactA* | TTTGAGACCTTCAACGCCCC | mRNA level quantification of the gene *bcactA* |
| Rv*bcactA* | ACGTGAGTAACTCCGTCACC | mRNA level quantification of the gene *bcactA* |
| Fw*bctub* | TCCTTTCGGTCAACTCTTCCG | mRNA level quantification of the gene *bcβtub* |
| Rv*bctub* | CACCCTCAGTGTTAATGACCC | mRNA level quantification of the gene *bcβtub* |

distributed, the non-parametric Mann-Whitney U test was applied. All experiments were performed with the number of biological replicates indicated in the respective method descriptions. Data are presented as mean ± standard deviation. A *p*-value of < 0.05 was considered statistically significant.

## Results

### *Botrytis pseudocinerea* VD165 exhibits enhanced vegetative growth and superior stress resilience

To establish a baseline physiological comparison, the vegetative growth and stress tolerance of *B. pseudocinerea* VD165 were assessed against the reference *B. cinerea* strain B05.10 [8]. On solid YGG-agar medium, *B. pseudocinerea* VD165 demonstrated a significantly more vigorous growth phenotype. Its colonies expanded at an average rate of 1.7 cm/day, compared to 1.5 cm/day for *B. cinerea* B05.10. This resulted in a colony radius that was approximately 20% larger after three days of incubation (Fig 1A). This inherent capacity for rapid proliferation was also evident in liquid culture, where VD165 produced 25% more fresh weight and, more importantly, accumulated approximately 15% greater dry biomass than B05.10 after four days, a statistically significant difference (p < 0.05) that points to higher metabolic efficiency (Fig 1B).

Beyond its rapid growth, *B. pseudocinerea* VD165 displayed a markedly enhanced resilience to key stress factors. When challenged with oxidative stress induced by 5 mM of $H_2O_2$, the growth of B05.10 was severely restricted, showing 70% inhibition. In stark contrast, VD165 was highly tolerant, with its growth inhibited by only 20% (Fig 1C). The most striking difference was observed under conditions of cell wall stress. *B. pseudocinerea* VD165 showed significantly higher tolerance to the disruptive action of the detergent SDS, suggesting a more resilient cell wall architecture compared to B05.10. Both species exhibited similar levels of sensitivity to osmotic stress induced by 1.4 M of sorbitol (Fig 1C). Collectively, these data establish *B. pseudocinerea* not as a weaker ecological counterpart, but as a physiologically vigorous saprophyte with a superior intrinsic growth capacity and strong defense mechanisms against common oxidative and cell-membrane stresses.

### *B. pseudocinerea* VD165 displays a distinctive biochemical profile of lower ROS production and higher acidification

To investigate the underlying biochemical factors that might contribute to virulence, we compared the production of reactive oxygen species (ROS) and the capacity for medium acidification between the two species. Quantitative analysis using a DAB assay revealed a key difference in their oxidative physiology. *B. pseudocinerea* VD165 was found to produce approximately 25% less endogenous ROS compared to the reference strain *B. cinerea* B05.10 (Fig 2A). This lower baseline level of ROS production may contribute to its enhanced tolerance to external oxidative stress, suggesting a more tightly regulated redox homeostasis.

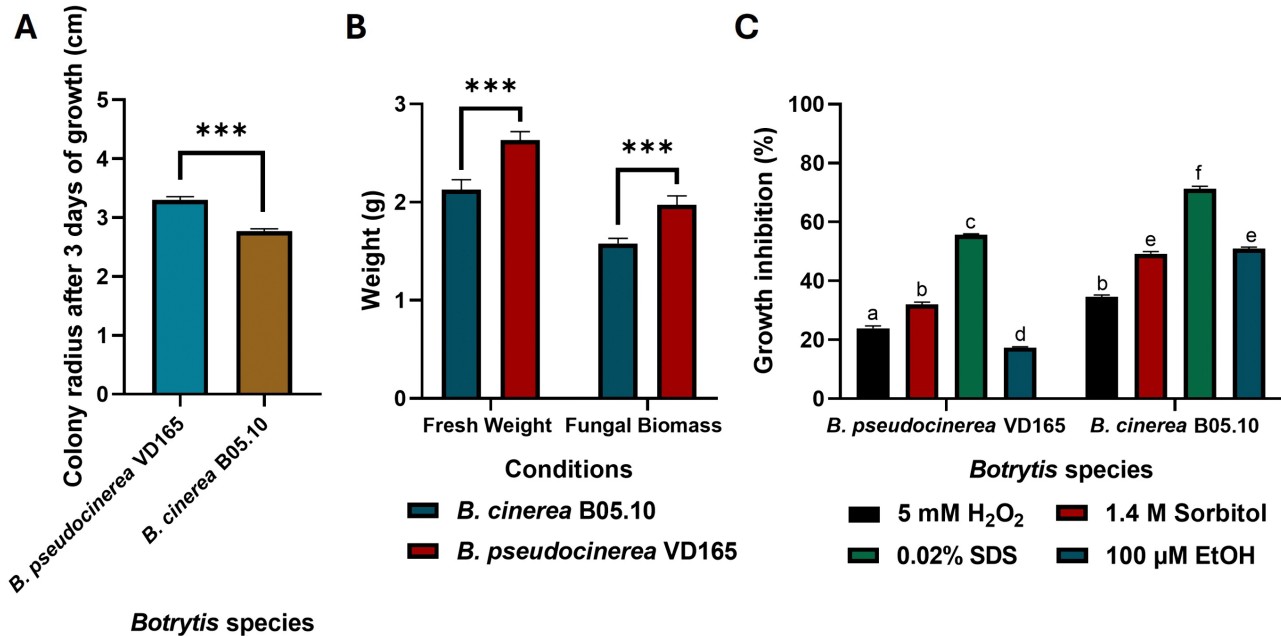

**Fig 1. Comparative physiology of *B. pseudocinerea* VD165 and *B. cinerea* B05.10.** **(A)** Colony radius (cm) over three days on YGG-agar. **(B)** Fresh and dry biomass (g) after four days in liquid YGG culture. **(C)** Growth inhibition (%) under oxidative (5 mM $H_2O_2$), cell wall (0.02% SDS), and osmotic (1.4 M Sorbitol) stress. * in **(A)** and **(B)** indicates significant differences between species ($p$-value < 0.05). Different letters above the bars in **(C)** mean that there are significant differences between all the conditions for both species ($p$-value < 0.05).

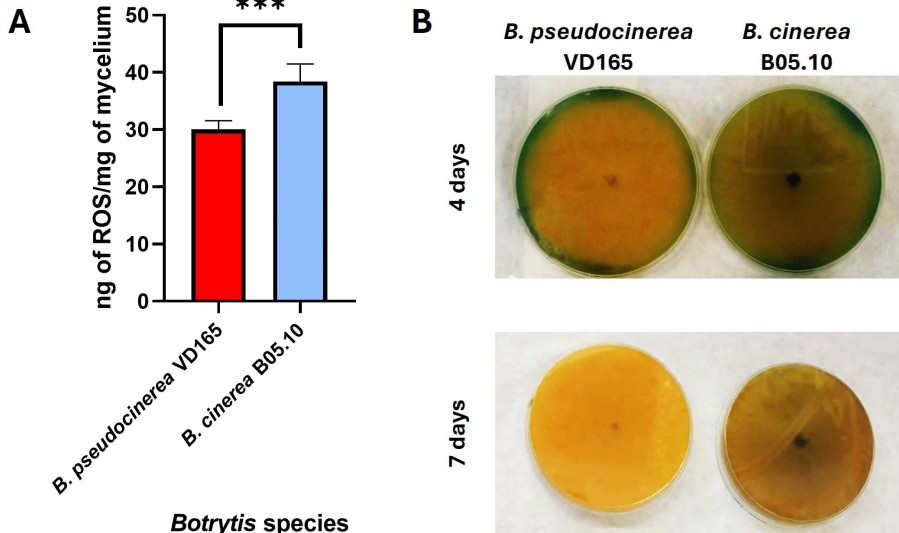

**Fig 2. Distinctive biochemical profiles of *B. pseudocinerea* VD165 and *B. cinerea* B05.10.** **(A)** Relative endogenous Reactive Oxygen Species (ROS) production measured by DAB assay. **(B)** Medium acidification capacity demonstrated by the colour change of bromocresol green indicator on malt-agar plates after 5 and 7 days of incubation. * in **(A)** indicates significant differences between species ($p$-value < 0.05).

Conversely, when cultured on malt-agar containing the pH indicator bromocresol green, *B. pseudocinerea* VD165 demonstrated a dramatically superior capacity for acidifying its environment. *B. pseudocinerea* VD165 produced a large, bright yellow halo indicative of a significant drop in pH, which was far more pronounced and developed more rapidly than the moderate acidification caused by B05.10, especially after 5 and 7 days of incubation (Fig 2B). This strong acidification is a characteristic of the secretion of organic acids, such as the well-established virulence factor oxalic acid, a well-established virulence factor in necrotrophic fungi that facilitates the degradation of host tissues. These opposing biochemical traits suggest a fundamental divergence in pathogenic strategy: *B. pseudocinerea* appears to rely less on a ROS-based oxidative burst for killing host cells and more on a potent, acid-driven mechanism to macerate and colonise host tissue.

## The pathogenicity of *B. pseudocinerea* VD165 is strongly host-dependent

To determine if the observed physiological and biochemical differences translate to pathogenic potential, virulence was assessed on a range of detached plant tissues. The results revealed a striking pattern of host-dependent pathogenicity. On gerbera petals, both *B. pseudocinerea* VD165 and *B. cinerea* B05.10 exhibited similar levels of infectivity, producing necrotic lesions of comparable size (Fig 3A and 3D). This indicates that on certain hosts, their virulence is equivalent.

However, a dramatic difference emerged on solanaceous hosts. On detached leaves of both tobacco and tomato, *B. pseudocinerea* VD165 proved to be significantly more virulent than its counterpart. It caused rapidly expanding water-soaked necrotic lesions whose diameters were, on average, 47% larger on tobacco and 56% larger on tomato compared to those induced by B05.10 at 72 hpi (Fig 3B-3D). This heightened aggression was also observed on tomato fruits, where a semi-quantitative assessment showed that approximately 30% of fruits inoculated with VD165 reached the final and most severe stage of infection (stage 4) by the end of the experiment, compared to only about 10% for B05.10 (Fig 3E). This host-specific enhancement was inverted on grapes. On this host, the canonical pathogen *B. cinerea* B05.10 was significantly more aggressive particularly in the later stages of infection progression (Fig 3F). Taken together, these data (Fig 3A-F) do not show a pattern of general weakness, but rather a clear evolutionary trade-off: *B. pseudocinerea* has enhanced virulence on Solanaceous hosts at the cost of reduced virulence on hosts like grape, a clear sign of host specialization. These results clearly demonstrate that the pathogenicity of *B. pseudocinerea* is not universally superior, but is highly dependent on the host substrate, pointing towards a specialised profile of phytotoxins that are particularly effective against certain plant families.

## The botcinin biosynthetic pathway is upregulated in *B. pseudocinerea* VD165

To investigate the molecular basis of a potentially unique chemical profile, we performed qRT-PCR analysis to quantify the expression of key secondary metabolite biosynthetic genes. We focused on *Bcboa6* and *Bcboa9*, the two polyketide synthase (PKS) genes required for the biosynthesis of the botcinin family of toxins [18]. The analysis revealed a dramatic and consistent upregulation of both genes in *B. pseudocinerea* VD165 compared to *B. cinerea* B05.10 across all tested conditions (Fig 4).

In axenic liquid culture, the expression levels of *Bcboa6* and *Bcboa9* in VD165 were already higher than in B05.10, with expression peaking at 14 days. This indicates that the botcinin pathway is constitutively "primed" for high-level production in *B. pseudocinerea*. Crucially, this high basal expression was further amplified during pathogenesis on its preferred hosts. At 48 hours post-inoculation on both tobacco and tomato leaves, the hosts where it displayed the highest virulence, the expression of *Bcboa6* and *Bcboa9* in VD165 surged to high levels far exceeding those detected in B05.10 under the same conditions. The transcriptional levels of *Bcboa6* and *Bcboa9* in *B. pseudocinerea* VD165 were not merely higher, but orders of magnitude greater than in *B. cinerea* B05.10 (Fig 4). This drastic overexpression mirrors the compensatory upregulation observed in *B. cinerea* Δ*Bcbot2* mutants, suggesting that the regulatory release of the botcinin pathway is a direct consequence of the botrydial pathway's loss [18]. In all instances, the induction of *Bcboa6* was consistently higher than that of *Bcboa9*. This strong, constitutive overexpression, which is significantly enhanced upon infection, provides

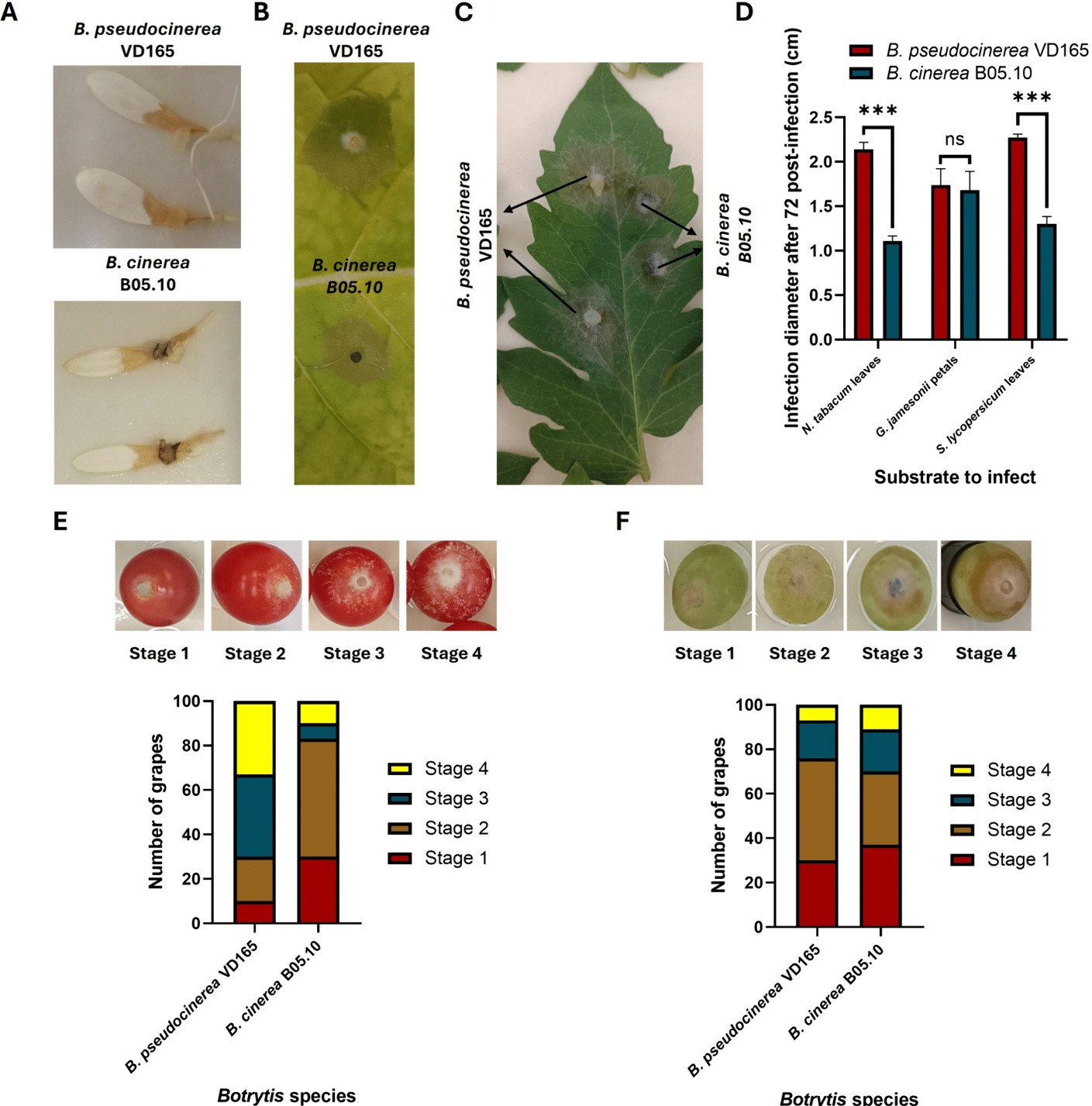

**Fig 3. Host-dependent pathogenicity of *B. pseudocinerea* VD165 and *B. cinerea* B05.10. (A)** Necrotic lesions on gerbera petals. **(B)** Necrotic lesions on tobacco leaves. **(C)** Necrotic lesions on tomato leaves. **(D)** Quantification of lesion diameters (cm) at 72 hpi. **(E)** Semi-quantitative assessment of infection progression on tomato fruits. **(F)** Semi-quantitative assessment of infection progression on grape fruits. ** and *** in **(D)** indicates significant differences between species ($p$-value < 0.01 and $p$-value < 0.001 respectively).

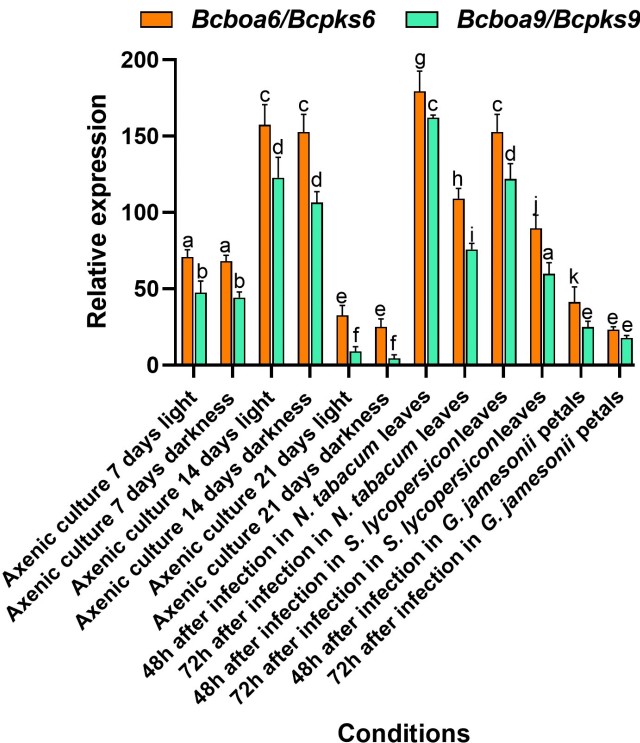

**Conditions**

**Fig 4. Relative expression of botcinin biosynthetic genes (*Bcboa6* and *Bcboa9*) in *B. pseudocinerea* VD165 compared to *B. cinerea* B05.10.** Gene expression was measured by qRT-PCR in axenic culture at 7 and 14 days, and during infection of tobacco and tomato leaves at 48 hpi. Expression levels are shown as relative quantification (using the $2^{-\Delta\Delta Ct}$ method) compared to *B. cinerea* as the reference. Different letters above the bars mean that there are significant differences between all the conditions tested (*p*-value < 0.05).

a clear molecular rationale for a metabolic profile that is heavily skewed towards the synthesis and deployment of botcinin-family phytotoxins as a primary offensive strategy.

### *Botrytis pseudocinerea* VD165 produces botcinin-family polyketides but critically lacks the botrydial sesquiterpene pathway

This study marks the first successful chemical characterization of the secondary metabolome of *B. pseudocinerea*. A bio-guided fractionation approach [19], using an onion cell plasmolysis assay to track phytotoxic activity, identified the extract from a 14-day culture grown under continuous light as the most potent. Large-scale fermentation under these conditions yielded 590 mg of crude ethyl acetate extract. Subsequent chromatographic purification led to the isolation and structural elucidation of several known members of the botcinin family of polyketides [26,27]. These included botcinic acid (**1**, 5.3 mg), botcineric acid (**2**, 3.3 mg), and their cyclic derivatives botcinin A (**3**, 12.34 mg) and botcinin B (**4**, 13.36 mg) [26], as well as botcinin E (**5**, 2.3 mg) (Fig 5) [27].

Critically, and in stark contrast to the known profile of *B. cinerea*, exhaustive analysis of all chromatographic fractions failed to detect botrydial or any other botryane-type sesquiterpenes, which are key virulence factors of *B. cinerea* [28]. This negative finding was substantiated by a key piece of biochemical evidence: the isolation of a substantial quantity of mevalonolactone (**6**, 44.35 mg). Mevalonolactone is a key intermediate precursor in the universal terpene biosynthetic pathway. Its accumulation in significant amounts provides compelling evidence that this pathway is active in its early stages but is truncated or functionally blocked downstream, preventing the biosynthesis of more complex sesquiterpene

**Fig 5. Chemical structures of secondary metabolites isolated from *B. pseudocinerea* VD165 and *B. cinerea* B05.10.** Compounds from the botcinin family produced by *B. pseudocinerea* include botcinic acid (**1**), botcineric acid (**2**), botcinin A (**3**), and botcinin B (**4**), and botcinin E (**5**). Mevalonolactone (**6**) is the terpene precursor accumulated by *B. pseudocinerea*. Isosclerone (**7**) was also isolated.

end-products like botrydial. The isolation of the naphthalenone polyketide isosclerone (**7**, 3.21 mg) was also achieved [29]. The complete absence of the botrydial pathway, a "lost toxin," combined with the clear evidence of a highly active botcinin pathway, clearly defines the unique and divergent toxic secondary metabolite profile of *B. pseudocinerea*.

## Discussion

This study provides the first multi-faceted investigation into the biology, pathogenicity, and chemical weaponry of the cryptic fungus *B. pseudocinerea*, revealing significant physiological and metabolic differences that distinguish it from its well-known sibling species, *B. cinerea*. Our findings move beyond simple characterisation to establish *B. pseudocinerea* VD165 isolate studied as a well-defined and distinct pathogen whose unique evolutionary trajectory and pathogenic strategy are defined by a fundamental rewiring of its secondary metabolism. The data presented here resolve a standing ecological paradox [8] and provide a compelling model for how functional divergence can drive speciation in pathogenic fungi.

### *B. pseudocinerea* isolate VD165 is a physiologically distinct and robust organism

Our results compellingly demonstrate that *B. pseudocinerea* VD165 is a highly adapted and vigorous organism, challenging any notion that it might be a less-fit ecological variant of *B. cinerea*. Its superior rate of vegetative growth in both solid and liquid culture points to high metabolic efficiency and a strong saprophytic capacity, which is crucial for survival and inoculum buildup on plant debris in agricultural settings. More significantly, its enhanced tolerance to specific stress factors indicates fundamental adaptations at the cellular level. The marked resistance to the cell-wall-disrupting agent SDS strongly suggests a more resilient cell wall architecture, a feature that would not only confer greater environmental durability but also critically influence the initial interactions with host plant surfaces and their defense mechanisms. Similarly, its heightened tolerance to oxidative stress, coupled with lower endogenous ROS production, points to a highly efficient redox management system.

In filamentous fungi, the interplay between stress response pathways and secondary metabolism is deeply integrated; secondary metabolites are not mere byproducts but are often key players in adaptation and defense [30]. For instance, in

*Aspergillus flavus*, the transcriptome composition and production of secondary metabolites, such as aflatoxin, are influenced by oxidative stress and carbon metabolism [31]. The high-level shift from one metabolic sink (botrydial) to another (botcinins) in *B. pseudocinerea* could free up metabolic resources (e.g., NADPH, acetyl-CoA) that can be reallocated to primary metabolism and stress defense systems. This directly connects the physiological data to the central hypothesis of a metabolic schism, suggesting that the observed strong phenotype is a direct manifestation of the specific set of metabolites it has evolved to produce, creating an organism well-equipped for its ecological niche.

### A divergent arsenal of virulence factors: linking host-specificity to metabolite profile

The host-dependent pathogenicity observed in our assays can be directly linked to the unique chemical profile of *B. pseudocinerea*. The fungus exhibits a clear biochemical trade-off: it produces significantly less ROS but generates a much more acidic local environment compared to *B. cinerea*, likely through the overproduction of oxalic acid. Oxalic acid is a key virulence determinant in many necrotrophic pathogens, acting to lower the ambient pH, which in turn chelates cell wall-stabilising $Ca^{2+}$ ions and optimises the activity of fungal cell-wall-degrading enzymes, thereby facilitating tissue maceration and decay [32]. This suggests an alternative pathogenic strategy that de-emphasises the oxidative burst, a common weapon for host cell killing, in favour of a potent, acid-driven degradation of host tissues.

This phenomenon can be understood in the context of the "trade-off hypothesis" in virulence evolution, which posits that pathogens face trade-offs between different components of their fitness, such as replication rate and infection duration [33]. In this case, *B. pseudocinerea* has "traded" the broad-spectrum utility of botrydial for a more specialized but highly potent suite of botcinins and strong acid secretion (putatively oxalic acid). This trade-off results in higher fitness on hosts susceptible to this specific chemical combination (*Solanaceae*), at the cost of lower fitness on hosts where botrydial is more effective (grapes). We propose a model where this specific combination of high acid secretion (likely oxalic acid) and a barrage of botcinin-family toxins is synergistically effective against certain hosts, particularly those in the *Solanaceae* family like tomato and tobacco, where *B. pseudocinerea* displayed its highest virulence. Our model is strongly supported by recent co-evolutionary studies. Our observation that virulence is highest on *Solanaceae* (Fig 3) and correlates with a significant *Bcboa* upregulation (Fig 4) can be explained by a newly discovered host-pathogen interaction. Phytoalexins produced by *Solanaceae* plants (such as tomato and potato)... namely rishitin and capsidiol, have been recently identified as potent *inducers* of the botcinic acid (BOA) biosynthetic gene cluster in *Botrytis* [34]. This suggests a sophisticated pathogenic strategy: the host's own chemical defenses are being hijacked to trigger the pathogen's primary offensive weapon. While this response is inducible in *B. cinerea*, our data (Fig 4) show that *B. pseudocinerea* maintains its BOA pathway in a constitutively 'primed' and strong upregulated state. It arrives at the host *pre-adapted* for this chemical interaction, allowing it to overwhelm the host's defenses and explaining the observed host-specific hypervirulence. Conversely, the complete lack of botrydial, a potent toxin known to be a major virulence factor for *B. cinerea* on a broad range of hosts, including grapes and apples, likely explains the reduced fitness of *B. pseudocinerea* on those particular hosts [20]. Studies in *B. cinerea* have shown that botrydial and botcinic acid play redundant yet collectively crucial roles in achieving full virulence; mutants lacking both toxins are significantly less pathogenic than those lacking only one [18]. Therefore, the constitutional loss of an entire arm of this dual-toxin system in *B. pseudocinerea* is expected to have substantial and defining consequences for its pathogenic capabilities, restricting its efficacy to hosts that are particularly susceptible to its remaining, highly upregulated chemical weapons. This provides a direct, mechanism-based explanation for the observed patterns of host specificity (Table 2).

### The central insight: A metabolic and evolutionary schism defines *B. pseudocinerea*

The most significant finding of this work is the discovery of a fundamental metabolic schism that defines the pathogenic identity of this *B. pseudocinerea* isolate: the complete absence of the botrydial biosynthetic pathway coupled with the constitutive and infection-induced overexpression of the botcinin pathway. This represents a major rewiring of the fungus's

**Table 2. Comparative summary of key physiological, biochemical, and pathogenic characteristics of *Botrytis pseudocinerea* VD165 and *Botrytis cinerea* B05.10.**

| Characteristic | *Botrytis pseudocinerea* VD165 | *Botrytis cinerea* B05.10 |
|---|---|---|
| **Physiology (Saprophytic Fitness)** | | |
| Vegetative Growth Rate | High (1.7 cm/day) | Moderate (1.5 cm/day) |
| Tolerance to Oxidative Stress ($H_2O_2$) | High (20% inhibition) | Low (70% inhibition) |
| Tolerance to Cell Wall Stress (SDS) | High | Low |
| **Biochemistry (Virulence-Related)** | | |
| Relative ROS Production | Low | High (Baseline) |
| Medium Acidification (Inferred) | High | Moderate |
| **Secondary Metabolism (Toxin Profile)** | | |
| Sesquiterpene Pathway (Botrydial) | Absent | Active and primary |
| Polyketide Pathway (Botcinins) | Active and highly upregulated | Active (secondary) |
| **Metabolic Signature** | | |
| Key Accumulated Precursor | Mevalonolactone (Terpene precursor) | None detected |
| **Pathogenicity Profile** | | |
| Virulence on Grapes | Moderate | High |
| Virulence on Tomato/Tobacco | High | Moderate |

secondary metabolism since its divergence from a common ancestor shared with *B. cinerea*. This conclusion is supported by a combination of chemical, biochemical, and molecular data. First, no botryane sesquiterpenes were detected in any culture extract. Second, the accumulation of the upstream precursor, mevalonolactone, provides definitive biochemical proof of a blocked or truncated terpene pathway [35]. The accumulation of mevalonolactone (**6**) is of particular significance. In fungal terpene biosynthesis, the absence of a functional sesquiterpene cyclase leads to the accumulation of mevalonate, which spontaneously lactonizes. Thus, the isolation of **6** serves as a definitive biochemical proxy for the genetic disruption of the botrydial pathway downstream of mevalonate kinase, identical to the metabolic signature of engineered *B. cinerea* Δ*Bcbot2* deletion mutants [36]. To elucidate the specific genetic lesion responsible for this blockage, we performed an *in silico* analysis of the *B. pseudocinerea* reference genome (strain Bp36). This analysis confirmed the presence of the essential upstream mevalonate kinase (*ERG12*) but revealed the specific absence of the *Bcbot2* sesquiterpene cyclase gene. This deletion prevents the channeling of farnesyl diphosphate (FPP) into the botryane pathway, causing a metabolic overflow that results in the accumulation and excretion of mevalonolactone. This specific metabolic phenotype mirrors that of the *B. cinerea* strain B459, which similarly accumulates mevalonolactone when toxin biosynthesis is impeded [37], confirming that other putative sesquiterpene cyclases do not functionally replace the *Bcbot2* sink under these conditions. Third, the high-level upregulation of the *Bcboa* genes at the transcriptional level provides the molecular mechanism for the observed overproduction of botcinins.

Crucially, this study leverages *B. pseudocinerea* VD165 as a 'natural knockout' model to validate the regulation of fungal virulence factors. Previous genetic studies in *B. cinerea* established that the deletion of the sesquiterpene cyclase gene (*Bcbot2*) leads to a metabolic blockage, accumulation of mevalonolactone, and a compensatory upregulation of the botcinic acid gene cluster. Our data reveal that *B. pseudocinerea* has arrived at this exact phenotype through natural evolutionary processes. The accumulation of mevalonolactone (Fig 5) serves as definitive biochemical evidence of a truncated terpene pathway, while the strong overexpression of *Bcboa* genes (Fig 4) confirms the compensatory regulatory loop. This phenotypic convergence between the engineered Δ*Bcbot2* mutant and the wild *B. pseudocinerea* species provides robust functional validation of the 'metabolic schism' hypothesis without the need for artificial genetic manipulation in the recalcitrant *B. pseudocinerea* background [18,36].

The hypothesis that this metabolic schism is a species-defining trait rather than an isolate-specific anomaly is strongly supported by phylogenetic parsimony. *B. pseudocinerea* belongs to *Botrytis* Clade 1, a sister lineage to the *B. cinerea* complex (Clade 2). Significantly, *Botrytis fabae*, another member of Clade 1 and a strict host specialist, has also been shown to lack the botrydial biosynthetic cluster. The presence of the cluster in Clade 2 (*B. cinerea*) and its absence or degeneration in multiple Clade 1 species (*B. pseudocinerea*, *B. fabae*) suggests that the loss of botrydial production is a synapomorphy, a shared derived character, of the Clade 1 lineage. This phylogenetic context effectively shields the findings from the limitation of single-isolate analysis, as VD165 acts as a representative archetype for the evolutionary trajectory of its clade [11].

The characterization of this divergence as a 'loss' in the *B. pseudocinerea* lineage, rather than a 'gain' in *B. cinerea*, is strongly supported by phylogenetic parsimony. Phylogenetic analysis places *B. cinerea* (Clade 2) in a distinct sister lineage from the one containing both *B. pseudocinerea* and *B. fabae* (Clade 1) [11]. Notably, *B. fabae*, a host specialist and the closest-sequenced relative to *B. pseudocinerea*, has also been shown to have lost the entire botrydial biosynthetic cluster as part of a broader pattern of genomic decay and specialization [12]. The presence of the cluster in the outgroup (*B. cinerea*) and its absence in the two sister species of Clade 1 implies that the most parsimonious evolutionary scenario is a single loss event in the common ancestor of *B. pseudocinerea* and *B. fabae*. This loss, therefore, represents a key evolutionary divergence that set this lineage on a different pathogenic trajectory.

While a full comparative genomic analysis was outside the scope of this study, we acknowledge that differences beyond the botrydial and botcinin clusters undoubtedly exist. However, the divergence of secondary metabolite (SM) biosynthetic gene clusters (BGCs) is a primary and well-established driver of ecological adaptation and speciation in pathogenic fungi [38]. These BGCs frequently reside in labile, AT-rich, subtelomeric regions, making them 'hotspots' for rapid evolution, gene loss, and genomic decay. Indeed, both the botrydial cluster [39] and the botcinic acid cluster [40] in *B. cinerea* are located in such unstable genomic regions. Therefore, while other genetic variations contribute, the complete functional loss of one major toxin pathway (or BGC) and the compensatory upregulation of another represents a striking and defining shift in pathogenic weaponry, and is the most parsimonious explanation for the divergent pathogenic lifestyles observed.

This process of "evolution by subtraction" is not unique to *Botrytis* and serves as a powerful example of how the loss of key functions, far from being solely detrimental, can facilitate the evolution of alternative, specialized pathogenic lifestyles [21]. To place our findings in the broader context of fungal pathogen evolution, we compiled a comparative analysis of different evolutionary mechanisms across several key genera (Table 3).

As illustrated in Table 3, the evolutionary trajectory of *B. pseudocinerea* is a clear example of how the loss of a biosynthetic gene cluster (BGC) can be a defining event. In *B. fabae*, a close relative, the loss of the same botrydial cluster is part of a broader pattern of gene decay that leads to strict host specialization [12]. This provides a powerful parallel within

Table 3. Comparative analysis of the evolution of secondary metabolite pathways in fungal pathogens.

| Pathogen Genus | Key Species Example(s) | Primary Evolutionary Mechanism | Key Metabolic Pathway(s) Affected | Impact on Pathogenicity/ Host Range | Key References |
|---|---|---|---|---|---|
| *Botrytis* | *B. pseudocinerea*/ *B. fabae* | BGC Loss & Compensatory Upregulation | Loss of Botrydial; Upregulation of Botcinins | Defines new pathogenic identity and ecological niche; specialization. | This study; [12] |
| *Fusarium* | *F. oxysporum*/ *F. graminearum* | BGC Presence/Absence Polymorphism | Depudecin, Gibberellins | Differentiates pathogenic vs. non-pathogenic lifestyles; host-specific adaptations. | [31,41,42] |
| *Aspergillus* | *A. flavus*/ *A. fumigatus* | Pan-metabolome Diversity; BGC Divergence | Aflatoxin, Fumagillin/ Pseurotin | Defines population-specific ecology; differentiates pathogens from non-pathogenic relatives. | [31,43,44] |
| *Metarhizium* | *M. anisopliae*/ *M. robertsii* | Gene Family Expansion & HGT | Expansion of PHI genes, GPCRs | Drives evolution from specialists to generalists (broader host range). | [45,46] |

the same genus. In *Fusarium*, the presence or absence of specific BGCs, such as that for depudecin, directly distinguishes pathogenic strains from biocontrol strains, demonstrating that BGC content is a key determinant of ecological function [31]. Similarly, in *Aspergillus*, intraspecific variation in BGCs correlates with population ecology, and differences in virulence-associated BGCs (e.g., fumagillin) distinguish pathogens like *A. fumigatus* from their non-pathogenic relatives [31,43]. All these examples reinforce the idea that the BGC profile is fundamental to a fungus's identity. In stark contrast, the "evolution by subtraction" model observed in *Botrytis* opposes the trajectory seen in the entomopathogenic fungus *Metarhizium*, where the evolution towards a broader host range (generalization) was driven by the *expansion* of gene families and horizontal gene transfer, not by gene loss [45,46]. This comparison is crucial, as it demonstrates that there is no single path to pathogenic success; the evolutionary trajectory of *B. pseudocinerea* represents one of several valid strategies.

## Open questions and future perspectives

Although this study provides a deep characterisation, it is based on a single representative isolate for each species and opens several fundamental questions for future research.

First, what selective pressures drove this metabolic schism? It is possible that the common ancestor inhabited a niche where botrydial was less effective, or that a chance mutation disabling the botrydial pathway allowed the invasion of a new niche (e.g., *Solanaceae*) where botcinins are more potent. Second, what is the energetic trade-off? The loss of the complex botrydial pathway could free up significant metabolic resources (acetyl-CoA, ATP, NADPH), contributing to the faster growth and greater stress tolerance observed in *B. pseudocinerea*. Third, is the loss irreversible? Given the evidence of genomic decay at the botrydial locus in related species [12], it is likely that the pathway has been permanently lost, although the possibility of re-acquisition through horizontal gene transfer, a known mechanism for BGC evolution, cannot be entirely dismissed. Finally, how does the host respond? It is crucial to investigate whether tomato and tobacco plants possess specific susceptibility factors that are targeted by botcinins but not by botrydial, which would explain the enhanced virulence.

Future investigations should expand to include population-level genomic and metabolomic analyses to confirm that the "lost toxin" profile is a conserved, defining trait across the *B. pseudocinerea* species [47]. The generation of knockout mutants of the *Bcboa* genes in the *B. pseudocinerea* background is necessary to definitively confirm the essential role of botcinins in its specialized virulence on tomato and tobacco.

## Supporting information

**S1 File. Minimal data set underlying the findings and figures.**
(XLSX)

## Acknowledgments

Use of NMR facilities at the Servicio Centralizado de Ciencia y Tecnología (SCCYT) of the University of Cádiz is acknowledged. The authors wish to thank Dr. M. Viaud and Dr. A.S. Walker for generously providing the *Botrytis pseudocinerea* strain for this research.

## Author contributions

**Conceptualization:** Victor Coca-Ruiz, Isidro G. Collado.

**Data curation:** Victor Coca-Ruiz.

**Formal analysis:** Victor Coca-Ruiz.

**Funding acquisition:** Josefina Aleu, Isidro G. Collado.

**Investigation:** Victor Coca-Ruiz, Adrián García-Barba.

**Methodology:** Victor Coca-Ruiz, Josefina Aleu.

**Project administration:** Isidro G. Collado.

**Supervision:** Josefina Aleu, Isidro G. Collado.

**Visualization:** Victor Coca-Ruiz.

**Writing – original draft:** Victor Coca-Ruiz, Josefina Aleu, Isidro G. Collado.

**Writing – review & editing:** Josefina Aleu, Isidro G. Collado.

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
