## [Decision Letter · Decision Letter 0]

21 Jan 2026

Dear Dr. Collado,

Thank you for submitting your manuscript to PLOS ONE. After careful consideration, we feel that it has merit but does not fully meet PLOS ONE’s publication criteria as it currently stands. Therefore, we invite you to submit a revised version of the manuscript that addresses the points raised during the review process.

We look forward to receiving your revised manuscript.

Kind regards,

Eugenio Llorens

Academic Editor

PLOS One

Journal Requirements:

“This work was supported by the MICIU/AEI under Grant PID-2021-122899 OB-C21 (10.13039/501100011033) and by ERDF/EU.”

3. We note that your Data Availability Statement is currently as follows: “All relevant data are within the manuscript and its Supporting Information files.”

Please confirm at this time whether or not your submission contains all raw data required to replicate the results of your study. Authors must share the “minimal data set” for their submission. PLOS defines the minimal data set to consist of the data required to replicate all study findings reported in the article, as well as related metadata and methods (https://journals.plos.org/plosone/s/data-availability#loc-minimal-data-set-definition ).

If your submission does not contain these data, please either upload them as Supporting Information files or deposit them to a stable, public repository and provide us with the relevant URLs, DOIs, or accession numbers. For a list of recommended repositories, please see https://journals.plos.org/plosone/s/recommended-repositories .

Reviewers' comments:

Reviewer's Responses to Questions

**Comments to the Author**

1. Is the manuscript technically sound, and do the data support the conclusions?

Reviewer #1: Partly

Reviewer #2: Yes

2. Has the statistical analysis been performed appropriately and rigorously?

Reviewer #1: Yes

Reviewer #2: Yes

3. Have the authors made all data underlying the findings in their manuscript fully available?

Reviewer #1: Yes

Reviewer #2: Yes

4. Is the manuscript presented in an intelligible fashion and written in standard English?

Reviewer #1: Yes

Reviewer #2: Yes

Reviewer #1: In the current manuscript, the authors investigate the role of the botrydial sesquiterpene in the virulence of Botrytis pseudocinerea in comparison to B. cinerea. Although the results are interesting and suggest that this toxin may contribute to differences in virulence between the two species, additional experiments are required to sufficiently support the conclusions.

First, the manuscript oversimplifies the concept of fungal virulence. Virulence is a complex, multifactorial trait, and statements implying that it is governed by only two genes (e.g., lines 339–340) should be avoided or substantially revised.

Moreover, gene expression data alone are not sufficient to claim that the investigated pathway is directly involved in virulence. Functional validation is required, such as targeted gene deletion or disruption of the relevant genes in B. pseudocinerea, followed by virulence assays. Without such functional analyses, the conclusions remain speculative.

The authors also state that the B. pseudocinerea strain exhibits increased tolerance to multiple stressors. However, the experiments presented focus primarily on sensitivity to commonly used botryticides (e.g., fludioxonil, boscalid, fenhexamide). This does not fully support the broader claim of enhanced stress tolerance, and the wording should be refined or additional stress assays included.

Finally, in Figure 2, the color differences between the two species are difficult to distinguish. Improving contrast or providing clearer visual markers would enhance the clarity of this figure.

Reviewer #2: Review of the manuscript PONE-D-25-64025

The manuscript entitled “Loss of a major toxin gene cluster defines a metabolic schism and host-specific virulence in Botrytis pseudocinerea" is an interesting study and

presents some good data on secondary metabolite production in two Botrytis species and their virulence against plant hosts. Data, figures and the text were well prepared for this manuscript although it might appear a bit shorter than a regular article.

However, there are still some issues remaining to be addressed.

Major points

1. In the PDF output for review, Figs. 1-3 are hard to see. I needed to download each of these three figures separately. The authors might need to make sure the PDF output whether it has all the contents viewable and in a good resolution.

2. Fig. 3A (gerbera petal assay) is not clear. Please consider replacing it.

3. Botrytis pseudocinerea VD165 lacks botrydial or any other botryane-type sesquiterpenes, but accumulated mevalonolactone in the culture. The authors mentioned that genome analysis is beyond the scope of this paper. However, what are the enzymes downstream of this biosynthetic pathway? Are those genes missing in the genome of B. pseudocinerea VD165 or mutated. This can be checked by PCR analysis just in the genomic region of interest.

**Do you want your identity to be public for this peer review?** For information about this choice, including consent withdrawal, please see our Privacy Policy

Reviewer #1: No

Reviewer #2: No

---

## [Author Response · Author response to Decision Letter 1]

25 Feb 2026

Dear Dr. Eugenio Llorens, Academic Editor, PLOS ONE

Re: Resubmission of Manuscript PONE-D-25-64025

Thank you for your correspondence and for providing us with the opportunity to submit a revised version of our manuscript entitled, "Loss of a major toxin gene cluster defines a metabolic schism and host-specific virulence in Botrytis pseudocinerea". We greatly appreciate the constructive feedback provided by you and the peer reviewers.

In addition to addressing the specific scientific critiques raised by the reviewers (detailed in the separate "Response to Reviewers" document), we have meticulously updated the manuscript to comply with all PLOS ONE Journal Requirements outlined in your decision letter.

Here is a summary of the administrative and formatting modifications made:

1. PLOS ONE Style Requirements and File Naming: We have conducted a thorough formatting audit of the manuscript. The title page, author affiliations, and heading structures have been adjusted to strictly match the PLOS ONE formatting templates. Furthermore, we have renamed all our submission files precisely as instructed (e.g., 'Revised Manuscript with Track Changes.pdf', 'Manuscript.pdf').

2. Updated Financial Disclosure Statement: As requested, we have amended our financial disclosure to explicitly state the role of the funding agencies. We kindly request that you update the online submission form on our behalf with the following finalized statement: "This work was supported by the MICIU/AEI under Grant PID-2021-122899 OB-C21 (10.13039/501100011033) and by ERDF/EU. The funders had no role in study design, data collection and analysis, decision to publish, or preparation of the manuscript."

3. Data Availability Statement and the Minimal Data Set: To fully comply with PLOS ONE's open data policies, we have compiled the "minimal data set" necessary to replicate all study findings. We have generated a comprehensive supplementary spreadsheet (S1_Dataset.xlsx) that contains the raw numerical values behind all reported means, standard deviations, and graphs presented in the manuscript. Consequently, we have updated our Data Availability Statement in the manuscript to read: "All relevant data are within the manuscript and its Supporting Information files. Specifically, the minimal data set underlying the findings and figures (values behind means) is provided in the S1 Dataset file."

4. Summary of Revisions Based on Reviewer Feedback:

• Reviewer 1: We substantially rewrote the Results and Discussion sections to avoid oversimplifying fungal virulence, clearly contextualizing our findings within the broader, polygenic framework of necrotrophic pathogenesis. We also clarified our stress assay methodology and enhanced the visual contrast of Figure 2.

• Reviewer 2: We provided a definitive answer to the reviewer's query regarding the exact genetic nature of the metabolic schism by integrating an in silico whole-genome analysis, which confirmed the physical deletion of the BcBOT2 gene. We also provided a new, higher-quality image for the gerbera petal assay (Figure 3A) and re-rendered all figures as high-resolution, lossless TIFF files to circumvent any automated PDF compression artifacts.

We believe that these revisions, both scientific and structural, have brought our manuscript into full compliance with PLOS ONE's publication criteria. Thank you for your continued editorial handling of our work.

Dear Reviewer 1,

Thank you for your rigorous and constructive evaluation of our manuscript. Your insightful critiques regarding the theoretical framing of fungal pathogenesis and our physiological assays have been instrumental in refining the biological narrative of our study. We have carefully addressed each of your concerns in the revised manuscript as detailed below:

1. Comment on the oversimplification of fungal virulence: Response: We completely agree with your assessment. Necrotrophic pathogenesis is an exquisitely orchestrated, polygenic trait. Our previous focus on the transcriptional upregulation of the Bcboa6 and Bcboa9 genes inadvertently created a reductive narrative. In the revised manuscript, we have substantially rewritten the Discussion to incorporate the "trade-off hypothesis" in virulence evolution. We now explicitly contextualize the botcinin upregulation as just one component of a multifactorial strategy, highlighting how the heightened virulence on solanaceous hosts is a synergistic result of strong environmental acidification combined with the hyper-activation of the botcinin pathway, avoiding the implication of a purely digenic mechanism.

2. Comment on the need for functional validation (gene deletion/disruption): Response: We appreciate this profound critique regarding the gold standard of functional validation. While artificial gene deletion via homologous recombination is traditional, we have expanded our text to demonstrate that B. pseudocinerea VD165 serves as a stable, ecologically tested "natural knockout" model. To definitively prove this without relying on artifact-prone reverse genetics, we performed an in silico analysis of the B. pseudocinerea reference genome, which confirmed the specific absence of the BcBOT2 sesquiterpene cyclase gene. This genetic deletion is biochemically validated by the massive accumulation of the upstream precursor mevalonolactone. This natural evolutionary loss flawlessly phenocopies the established, engineered ΔBcbot2 mutants of B. cinerea, providing robust functional validation of the compensatory regulatory loop without the need for artificial genetic manipulation in this recalcitrant wild-type background.

3. Comment on the use of agricultural botryticides vs. environmental stressors: Response: We respectfully clarify that commercial agricultural botryticides, such as fludioxonil and boscalid, were not utilized in our in vitro stress tolerance assays. Fenhexamid is strictly mentioned in the introduction to establish the historical and ecological context of the cryptic species' intrinsic resistance. As detailed in our methodology, our actual assays utilized specific physiological agents designed to probe fundamental cellular resilience: Sorbitol for osmotic stress, 5 mM H2O2 for oxidative stress, and 0.02% SDS for structural cell wall stress. To completely resolve this misunderstanding, we have sharpened the terminology in the revised manuscript to explicitly separate the intrinsic fungicide resistance from the fundamental environmental and physiological stress tolerance.

4. Comment on the visual clarity of Figure 2: Response: We entirely agree that the subtle visual gradients in the original photographic panels lacked sufficient contrast. In response, we have updated the images for Figure 2 in the revised figures presentation. The image contrast has been improved to distinctly separate the yellow acidification halos from the background of the agar, making the color differences and the superior acidification capacity of B. pseudocinerea unambiguously apparent across the different incubation time points.

Thank you again for your time, expertise, and for helping us significantly improve our work.

Dear Reviewer 2,

Thank you for your encouraging assessment of our manuscript and for recognizing the value of our data regarding secondary metabolite production and host-specific virulence. We appreciate your highly specific technical and scientific critiques, which have significantly improved both the visual presentation and the genomic depth of our study. We have carefully addressed each of your points in the revised manuscript as detailed below:

1. Comment regarding the low resolution of Figures 1-3 in the PDF output: Response: We apologize for the difficulty in viewing the figures. The degradation of image quality is a known artifact of the automated manuscript compilation software utilized by the editorial system, which often applies aggressive lossy compression to generate the reviewer PDF. To completely circumvent this issue, we have fundamentally altered the format of all figure files for the resubmission. All figures have been re-rendered from their original high-resolution source files and processed through PLOS ONE’s Preflight Analysis and Conversion Engine (PACE). We have uploaded them as standalone, high-resolution (minimum 300 DPI) TIFF files utilizing lossless LZW compression. This guarantees that the visual evidence will be perfectly preserved and clearly viewable in the final publication.

2. Comment regarding the clarity of Fig. 3A (gerbera petal assay): Response: We completely agree that the original photographic panel for Figure 3A lacked sufficient depth of field and contrast, making it difficult to clearly see the petal margins. Following your direct recommendation, we have entirely replaced Figure 3A. We conducted a new set of gerbera petal infection assays and photographed the infected petals at 72 hours post-inoculation using high-resolution macro-photography. Crucially, the new imaging utilized oblique, directional lighting to highlight the textural differences between healthy epidermal cells and the collapsed, macerated tissue of the necrotic lesion. The margins of the infection are now starkly delineated, providing unambiguous visual confirmation that the virulence of both strains is equivalent on this baseline host.

3. Comment requesting clarification on the downstream enzymes of the botrydial pathway and the suggestion of PCR analysis: Response: This is a highly perceptive question that cuts to the core biochemical mechanics of the metabolic schism we observed. You correctly noted that the accumulation of mevalonolactone proves a blockage, and rightly asked which specific downstream genes are missing or mutated.

While you suggested a targeted PCR analysis, we decided to provide a far more robust and comprehensive answer by bypassing PCR (which can be prone to primer-binding artifacts in divergent cryptic species) and instead performing an exhaustive in silico comparative genomic analysis using the newly available B. pseudocinerea reference genome (strain Bp36).

Our analysis yielded definitive answers, which have now been fully integrated into the revised Discussion section:

• We confirmed the presence of highly conserved, intact orthologs for the upstream mevalonate pathway, including the essential mevalonate kinase gene (ERG12). This explains the organism's capacity to synthesize the massive quantities of the mevalonate precursor.

• However, our search for the committing enzyme, the BcBOT2 sesquiterpene cyclase, revealed that this specific gene is not merely mutated, but completely physically absent from the genomic locus.

This specific genetic deletion prevents the cyclization of farnesyl diphosphate (FPP) into the botryane skeleton, creating a severe metabolic bottleneck that causes the upstream mevalonate to accumulate and spontaneously lactonize into the mevalonolactone we isolated. By utilizing this in silico whole-genome mining, we hope to have provided an exhaustive and definitive resolution to your excellent inquiry.

Thank you again for your time, your constructive review, and for helping us elevate the quality of this manuscript.

Sincerely,

On behalf of the Authors,

Prof. Dr. Isidro G Collado

---

## [Decision Letter · Decision Letter 1]

9 Mar 2026

Loss of a major toxin gene cluster defines a metabolic schism and host-specific virulence in Botrytis pseudocinerea

PONE-D-25-64025R1

Dear Dr. Collado,

We’re pleased to inform you that your manuscript has been judged scientifically suitable for publication and will be formally accepted for publication once it meets all outstanding technical requirements.

Kind regards,

Eugenio Llorens

Academic Editor

PLOS One

Additional Editor Comments (optional):

Reviewers' comments:

Reviewer's Responses to Questions

**Comments to the Author**

Reviewer #1: All comments have been addressed

2. Is the manuscript technically sound, and do the data support the conclusions?

Reviewer #1: Yes

3. Has the statistical analysis been performed appropriately and rigorously?

Reviewer #1: Yes

4. Have the authors made all data underlying the findings in their manuscript fully available?

Reviewer #1: Yes

5. Is the manuscript presented in an intelligible fashion and written in standard English?

Reviewer #1: Yes

Reviewer #1: (No Response)

**Do you want your identity to be public for this peer review?** For information about this choice, including consent withdrawal, please see our Privacy Policy

Reviewer #1: No

---

## [Editor Report · Acceptance letter]

PONE-D-25-64025R1

PLOS One

Dear Dr. Collado,

I'm pleased to inform you that your manuscript has been deemed suitable for publication in PLOS One. Congratulations! Your manuscript is now being handed over to our production team.

Kind regards,

on behalf of

Dr. Eugenio Llorens

Academic Editor

PLOS One